# Expectations and Assumptions: Examining the Influence of Staff Culture on a Novel School-Based Intervention to Enable Risky Play for Children with Disabilities

**DOI:** 10.3390/ijerph18031008

**Published:** 2021-01-23

**Authors:** Patricia Grady-Dominguez, Jo Ragen, Julia Sterman, Grace Spencer, Paul Tranter, Michelle Villeneuve, Anita Bundy

**Affiliations:** 1Department of Occupational Therapy, Colorado State University, Fort Collins, CO 80528, USA; anita.bundy@colostate.edu; 2Faculty of Medicine and Health, University of Sydney, Camperdown, NSW 2006, Australia; jrag7001@uni.sydney.edu.au (J.R.); michelle.villeneuve@sydney.edu.au (M.V.); 3Department of Occupational Therapy, University of Minnesota, Minneapolis, MN 55455, USA; sterm154@umn.edu; 4Faculty of Health, Education, Medicine and Social Care, Anglia Ruskin University, Cambridge CB1 1PT, UK; grace.spencer@anglia.ac.uk; 5School of Science, University of New South Wales, Sydney, NSW 2052, Australia; p.tranter@adfa.edu.au

**Keywords:** disabilities, play, risky play, school culture, staff culture, teacher perceptions

## Abstract

Risky play is challenging, exciting play with the possibility of physical, social, or emotional harm. Through risky play, children learn, develop, and experience wellbeing. Children with disabilities have fewer opportunities than their typically developing peers to engage in this beneficial type of play. Our team designed a novel, school-based intervention to address this disparity; however, our intervention yielded unexpected quantitative results. In the present study, we qualitatively examined divergent results at two of the five schools that participated in the intervention. Specifically, we aimed to explore how staff culture (i.e., shared beliefs, values, and practices) influenced the intervention. To explore this relationship, we employed a retrospective, qualitative, multiple case study. We used thematic analysis of evaluative interviews with staff members to elucidate the cultures at each school. Then, we used cross-case analysis to understand the relationships between aspects of staff culture and the intervention’s implementation and results. We found that staff cultures around play, risk, disability influenced the way, and the extent to which, staff were willing to let go and allowed children to engage in risky play. Adults’ beliefs about the purpose of play and recess, as well as their expectations for children with disabilities, particularly influenced the intervention. Furthermore, when the assumptions of the intervention and the staff culture did not align, the intervention could not succeed. The results of this study highlight the importance of (1) evaluating each schools’ unique staff culture before implementing play-focused interventions and (2) tailoring interventions to meet the needs of individual schools.

## 1. Introduction

Risky play is challenging and exciting play with the possibility of physical injury, failure, or emotional harm [1,2]. Risky play affords opportunities for children to test their limits, explore boundaries, and manage real risks [1,3]. Kleppe and colleagues (2017) emphasized the ambiguity inherent in risky play, describing it as “play that involves uncertainty and exploration—bodily, emotional, perceptional or environmental—that could lead to either positive or negative consequences” (p. 9). Through child-led, unstructured play with elements of risk, children develop social-emotional and communication skills and experience autonomy and resilience.

In school, recess provides an opportunity for students of all abilities to experience the benefits of free and risky play. For this to be possible, school staff supervising the playground must step back and allow children to manage risky situations. Unfortunately, staff members may be hesitant to step back due to threats of liability, personal reservations about risky play, and cultural shifts towards risk aversion and surplus safety [4,5]. These concerns may be even more pronounced for staff responsible for children with disabilities, who are often perceived as vulnerable and in need of protection [6].

Paradoxically, children with neurodevelopmental disabilities, such as autism spectrum disorder (ASD) and intellectual disability (ID), may benefit more from access to free and risky play than their typically developing peers. First, risky play affords children with disabilities opportunities to make decisions in low-impact scenarios. There is growing consensus that children and adults with disabilities should be enabled to make decisions about their medical care, education, and daily lives [7]; therefore, practicing these skills in the context of play is essential. Second, children who take risks in play are often more physically active [8,9]. Children with disabilities such as ASD and ID are at a high risk of developing pediatric obesity [10,11]; access to active, risky play may help mitigate this concern. Finally, cooperative forms of risky play (e.g., rough-and-tumble play) provide children with disabilities opportunities to practice social skills such as giving and receiving social cues [12]. Children with ASD/ID often demonstrate delays in social skills [13]; therefore, opportunities to practice are critical. Given these benefits (and school personnel’s reluctance to allow risky play), there is a need for intervention to address adults’ risk aversion for children with disabilities.

### 1.1. A Novel Intervention to Promote Risky Play for Children with Disabilities

Our team developed a novel intervention to promote and shift adult attitudes around risky play in school programs for children with disabilities. The intervention comprised two components: one adult-directed and one child-directed. The adult-directed portion (Risk Reframing Workshops (RRWs)) involved workshops encouraging school staff and parents to discuss the benefits and challenges of risky play for their students. For the child-directed portion, we introduced loose, recycled materials to the playground (e.g., tires, tarps, and barrels) to encourage cooperative, active, risky play. For a full description of the intervention, see Bundy and colleagues (2015). Table 1 describes the quantitative outcome measures we collected as well as analysis processes.

Fundamental assumptions about play and disability underscored this intervention. Our team, led by an occupational therapist, approached play as the primary occupation of childhood [17]. We defined play as any activity that is done primarily for its own sake: play is intrinsically motivated, requires some degree of internal control, and allows players to suspend reality. Players frame play such that it is separate from reality [17]. Additionally, we assumed that recess was a time for play, and that school staff would be motivated to encourage children’s unstructured play during recess. We expected that this would be especially true for adults working with children with disabilities, who often have goals surrounding social and play skills.

### 1.2. Staff Culture: A Factor to Consider When Implementing a Novel Program

Through this intervention, we asked school staff to undergo a significant ideological shift: from relative risk-intolerance to risk-tolerance in play. Perhaps not surprisingly, intervention processes and outcomes varied immensely across the five participating schools: some schools took quickly to the new approach, and their students benefited based on quantitative outcomes, while others struggled to see the benefits of risky or child-directed play, and their students experienced fewer gains.

When an intervention does not produce expected results, it is critical for researchers and stakeholders to understand why. In this study, we ask the fundamental question: why did the intervention work at some schools, and not at others? We begin with a hypothesis: the variation among schools can be explained, at least in part, by differences in staff cultures that existed prior to the intervention. For the purpose of this article, we define staff culture as the shared beliefs, norms, values, and practices upheld by adult members of the school community [18,19]. While school culture includes both pupils and adults, for the purpose of this paper, we focus on the organizational culture defined and enacted by school staff and administrators.

Domitrovich and colleagues [18] stated that “[school] culture influences the way things are routinely done” (p. 12). Novel interventions (including ours) may aim to shape or replace aspects of existing staff cultures. Therefore, it is prudent to expect that aspects of the staff culture present prior to the novel program will influence the intervention process. For example, McIsaac and colleagues [20] conducted a scoping review of factors that impacted schools’ adoption of school nutrition policies. They found that staff beliefs, values, and norms related to food and healthy eating were important determinants of schools’ adoption of nutrition policies. Unfortunately, we did not identify any previous literature exploring the impact of staff culture on interventions for students with disabilities, nor interventions surrounding the provision of risky play.

As demonstrated by the McIsaac et al. [20] report, the areas of staff culture that influence implementation are often specific to the intervention; in their case, beliefs about food and nutrition influenced the use of school nutrition policies. For the present study, we hypothesized that beliefs and practices surrounding three key areas: (1) disability, (2) the purpose of play/recess, and (3) risk-taking, may have influenced intervention processes and subsequent outcomes.

Staff culture around disability may include shared beliefs about teaching and learning, shared expectations for children with disabilities, and norms for supporting/interacting with children with disabilities [21]. Most of the literature examining staff members’ approaches to children with disabilities has focused on inclusion practices. However, in 2018, 31.2% of students with disabilities in Australia (where this study was conducted) attended either specialized schools or segregated classrooms [22]. None of the current literature explores staff cultures around disability—and the subsequent impacts on novel approaches/interventions—in these specialized environments.

Staff culture surrounding play may also influence play-based interventions for children with disabilities. Play theorist Sutton-Smith [23] described seven “rhetorics”—or ideologies—used to explain or justify play. These rhetorics include play as progress, fate, power, identify, imaginary, self, or frivolous. “Play as progress” focuses on the developmental benefits of play—this rhetoric often dominates school environments [24]. Staff who believe that the purpose of play is to enable children’s development may struggle to relinquish control over play, aiming to ensure that the children play in a way that will help them grow.

Finally, culture surrounding risk could influence staff members’ desire and ability to enact this intervention. While recent studies in the literature have examined parents’ beliefs about risky play, relatively few have examined school personnel’s perspectives—and how these beliefs play out on the school playground. Research examining early childhood educators’ beliefs leads to a fundamental contradiction: while many teachers believe that risky play benefits children, they also feel responsible for keeping children safe from any potential harm [25].

Given the potential for these three areas of staff culture to influence the intervention, the purpose of the present study was to examine widely held beliefs, practices, and norms about disability, play/recess, and risk at schools that participated in the study. Specifically, we employed a multiple case study approach, using two participating schools as cases, to address two related research questions:What was the staff culture (i.e., shared beliefs, norms, values, and practices) surrounding disability, play/recess, and risk-taking that characterized each of the two schools?How did staff culture differentially influence how staff engaged with the intervention and, subsequently, the quantitative outcomes?

## 2. Methods

We employed a retrospective, multiple case study to explore the influence of staff culture on intervention processes and subsequent outcomes for the novel playground intervention. We drew upon Yin’s [26] multiple case study methodology (described below) to address our research questions.

### 2.1. Case Selection

We selected two of the five schools that participated in the intervention for inclusion in the present study. Before selecting the two cases, the first author, PG-D, engaged in a debriefing session with two members of the intervention team, authors AB and JR. During five video conferences (~6 h), we reflected on each of the five participating schools. Our semi-structured discussions centered around school characteristics such as size and type, school policies, interactions with school staff, and intervention processes. Based on these discussions, we selected the two schools included in this study.

Our choice of these two schools represents Yin’s [26] maximum variation sampling: we chose two schools with differing characteristics, intervention processes, and outcomes. We chose the school as the unit of analysis so that we could explore school contextual factors in relation to the intervention.

### 2.2. Case Descriptions

In the section that follows, we briefly describe the relevant contextual factors of the two schools included in this study. Additionally, we provide a brief overview of the intervention process and outcomes at each school. To maintain the schools’ confidentiality, we withheld some factors.

#### 2.2.1. School A

School A was a small, private school exclusively for children with ASD with a student body of approximately 80 students. The school housed an early learning program, a primary school program, and a secondary school program. The school was located in a suburb of Sydney, Australia. While our research team did not collect data about children’s socioeconomic status, many children likely came from local upper- or middle-class families. Although the school did not charge tuition, parents were expected to make contributions to the school’s fundraising efforts. Additionally, children who attended this school had to have a recent ASD diagnosis. Most children who attended the school presented with moderate to severe ASD; many were nonverbal and had moderate-severe ID.

Staff employed a multidisciplinary approach to ASD-specific education, with multiple therapists and teachers collaborating to meet academic and therapeutic goals. The low ratio of students to staff (approximately 3:1) allowed for individualized support. Staff frequently and systematically collected data about goal attainment, adjusting individual support plans accordingly. Professionals received regular training in positive behavioral support (PBS) to manage behavior. Family involvement was paramount to the school philosophy. In addition to involving parents in individualized education plans (IEPs), School A offered family-, parent-, and sibling-support groups.

Children at School A had access to a playground with abundant fixed and loose play equipment. The playground where the primary school students played was approximately 625 m^2^ (6725 ft^2^). The playground had few natural features, with the exception of two mature trees. The ground was primarily soft-fall material, with some synthetic and natural grass. Playground features included a nest swing, a tire swing, a sling swing, a slide, stairs, a tunnel, a rock wall, a sliding/climbing pole, a climbing wall, a spinning barrel, a balance beam, a flying fox, a trampoline, a ball pit, and a sand pit. The children also had access to loose play materials including balls and items for water play. Many adults roamed the playground, with at least one adult for every 2–3 children.

The loose parts materials arrived on the playground at School A in June of 2015. The children did not immediately approach the intervention materials; instead, most children continued playing with the abundant equipment on the playground. Staff rarely encouraged children to use the materials. The majority of the children never used the materials; the few who did engage generally repeated the same activities each day (e.g., one student climbed in and out of a barrel; some students used a large wheel box).

Staff members and parents attended RRWs together. Leadership did not attend the RRWs. Throughout the RRWs, staff and parents were generally engaged and talkative. They shared common concerns, expectations, and desires for the children.

Quantitative outcomes suggested that the children experienced few benefits from the intervention. We collected data about 46 of the students regularly on the playground (35 males, 11 females). Figure 1 displays the playground sophistication score over time. The mean sophistication score decreased slightly from pre-intervention to post-intervention (M_pre_ = −0.70, SD_pre_ = 0.82; M_post_ = −1.17, SD_post_ = 0.82). Observers coded children at play as *more engaged* or *very engaged* slightly less often (57.7% of control observations, 54.8% of intervention observations, χ^2^ = 2.64, *p* > 0.05, Φ = −0.03). Coping inventory scores did not change significantly (M_con_ = 3.08, SD_con_ = 0.52; M_int_ = 3.56, SD_int_ = 0.71, *t* (28) = 1.692, *p* = 0.10, d = 0.24).

#### 2.2.2. School B

School B was a mainstream government school with a student body of approximately 270 children, ages 4–12. The school was located in a socially and economically disadvantaged neighborhood and had a large population of recently immigrated families. More than 75% of students spoke a language other than English at home. School B had three specialized classrooms for children with disabilities: one specifically for children with ASD diagnoses and two for children with moderate-severe ID. Our data collection focused on students from these three classrooms, although all children had access to the materials.

Each special education classroom had one qualified special education teacher; several paraprofessionals supported the teachers. The special education program (often called the “support unit”) was relatively new, with two of the three support classrooms added less than a year prior to the start of the intervention. Children in the classrooms had mixed diagnoses; many had ASD and/or ID. Perhaps because of the variety of diagnoses, staff did not follow specific academic or therapeutic approaches. All children had individual education plans (IEPs). At the time of intervention, School B had recently adopted PBS for all children, with and without disabilities—however, staff were still in training to use this approach. Family involvement was somewhat limited in the special education program, perhaps due to language or socioeconomic barriers.

Children at School B had access to a large playground, approximately 12,700 m^2^. The play space was largely empty, with fenced-in grassy fields and concrete areas with painted markings for games such as four-square. With the exception of several mature trees, natural play areas were limited to grass. School B had no fixed play equipment. Children could bring out some play items, including balls and art materials. There were two soccer goals on a grassy field.

The staff maintained a recess ratio of approximately 30:1. All staff had rotating playground duty. Ball games were very popular; prior to the intervention, sports were the only organized play option. Most children with disabilities were integrated onto the main playground; several of these children had direct support from a paraprofessional. A small number of younger students with more severe disabilities played in a separate space—notably, these children were not included in the present study because they did not access the loose materials during recess.

The loose parts materials arrived on the playground at School B in May of 2016. Both mainstream children and children from the support unit rushed immediately for the materials. Staff expressed concerns about the number of children attempting to play at once. Consequently, the research team and school agreed to a rotating schedule wherein children from the support unit always could play with the materials and mainstream children rotated each day by grade. Children used the materials to enact imaginary play schemes, build towers, and create novel games.

Staff members and parents attended separate RRWs due to scheduling constraints. The deputy principal and principal attended sessions with both groups. Administrative leadership also held separate staff meetings to discuss and troubleshoot the intervention.

Quantitative outcomes suggested that the children enjoyed several benefits from the intervention. We collected data about 20 randomly selected children (2 females; 18 males). Figure 2 displays the playground sophistication score over time. The mean sophistication score increased from control to intervention phase (M_con_ = −1.28, SD_con_ = 0.83; M_int_ = −0.31, SD_int_ = 0.72). Observers coded children at play as more engaged or very engaged more often (57.7% of control observations, 71.8% of intervention observations, χ^2^ = 18.28, *p* < 0.001, Φ = 0.14). Coping inventory scores increased significantly (M_con_ = 3.08, SD_con_ = 0.69; M_int_ = 3.56, SD_int_ = 0.65, *t* (15) = 3.38, *p* = 0.03, d = 0.64).

### 2.3. Data Sources

Interview data collected with staff members from each school served as the primary source of data for the present study. These interviews were originally conducted and recorded by another member of the research team (JS) to serve as evaluative feedback about the intervention study.

JS recruited participants on a voluntary basis. At the conclusion of the intervention (the third RRW), participating staff were asked if they were interested in participating in further interviews. JS conducted individual, semi-structured interviews using an interview guide developed and approved by members of the research team. The interviews were transcribed verbatim by a professional service. While the interviews served as a primary data source, PG-D also reviewed context reports for each school generated by research students. These context reports detailed the physical and social environments on the playground at each school, as well as other school factors germane to the intervention. Finally, PG-D conducted in-depth discussions with authors AB and JR at several points during analysis and write-up of the case report. Perspectives from these researchers, who were deeply embedded in each school during the intervention, served to clarify and complete our interpretations of each case.

### 2.4. Interview Participants

Table 2 describes interview participants. JS interviewed a total of 14 participants: 5 from School A and 9 from School B. Participants engaged in interviews on a voluntary basis. JS attempted to obtain diverse perspectives by interviewing participants with a variety of roles and levels of school experience.

### 2.5. Data Analysis

Author PG-D conducted all data analyses using NVivo qualitative analysis software (Winsteps.com, Beaverton, Oregon, USA). She employed Yin’s [26] cross-case synthesis approach. In cross-case synthesis, the researcher aims first to identify within-case patterns, and then to examine these patterns for replication across cases (i.e., cross-case syntheses).

#### 2.5.1. Within-Case Analysis

To operationalize Yin’s analytical strategy, PG-D first analyzed each case separately. She conducted within-case thematic analysis of staff interviews to highlight shared beliefs and practices around disability, play/recess, and risk-taking, employing Braun and Clarke’s [27] six-step approach. For each school, she (1) familiarized herself with the data, reading through each interview several times, making general notes about her reactions, thoughts, and preliminary codes; (2) worked through each interview systematically, creating initial codes; (3) searched the codes for potential themes, returning often to the research questions to examine how codes might be combined; (4) reviewed, revised, and refined the themes, circling back to the interviews to capture missed data; (5) defined, delimited, and named six themes: three from each school; and (6) produced a narrative report of the thematic analyses.

#### 2.5.2. Cross-Case Synthesis

After completing within-case analyses, PG-D conducted cross-case syntheses. Cross-case syntheses aimed to determine if patterns within each case were replicable in the other case [26]. PG-D compared themes from each school, searching for patterns, commonalities, and differences. By comparing the results from each school, PG-D generated theoretical propositions about the relationship between staff culture and the novel playground intervention.

#### 2.5.3. Trustworthiness

We took several steps to enhance confidence that data were collected and analyzed rigorously [28]. Given the authors’ involvement with the intervention (as interventionists and data analysts), we faced threats to neutrality. Therefore, the first author, PG-D, who did not participate in intervention implementation, led the study. PG-D also engaged in peer debriefing with an external researcher who was not engaged with the study. The external researcher provided feedback on themes, generated potential rival explanations, and assisted PG-D in clarifying central arguments. Additionally, the external researcher categorized approximately 10% of the data into the six themes generated by PG-D. Initially, the two researchers reached approximately 85% agreement on classification. They engaged in consensus-building discussion until they achieved 100% agreement. Additionally, PG-D maintained (and frequently reviewed) an audit trail tracking the progress and iterations of this analysis.

## 3. Findings

In the section that follows, we present the results of the cross-case analyses. For each school, we present the results of thematic analysis, followed by theoretical propositions pertaining to the relationships between themes and intervention processes/outcomes at each school.

### 3.1. School A

Below, we present the results of the analyses related to School A.

#### 3.1.1. Disability Makes Our Children Different

Staff at School A often emphasized the ways that disability set their students apart from other children. ASD and disability permeated the interviews with all staff members. One teacher reflected on play with her students: “Working here, it’s really different to where play is anywhere else. Just trying to teach kids that don’t even get the basic foundations of interacting with others. Yeah, just a completely different experience here than in other settings” (Teacher A2).

In addition to emphasizing the differences between their students and typically developing children, staff were clear that these differences engendered limitations in what their students could and could not do or understand. One teacher voiced concerns over her students’ understanding of danger:


*Some of them don’t get the rules and they don’t understand the danger. A lot of the kids don’t have any understanding of the danger I feel, some do, but I think a lot of them are just trying to get through the day. (Teacher A1)*


Another teacher explained why she felt that many of her students could not play cooperatively:


*Because it’s really tricky for our kids, like they’re overcoming all of their issues to be able to just play with us. To overcome all of their issues and then have to overcome all the other kids with autism’s issues, makes an interaction between two of our kids really quite difficult. (Teacher A4)*


Staff members at School A had considerable expertise in working with children with ASD and received frequent ASD-specific training. Their extensive knowledge about ASD may have led them to foreground disability during our interviews.

#### 3.1.2. The Playground Is a Pedagogical Place

School A had an established, shared philosophy around play. Throughout the interviews, staff emphasized that they used recess as a pedagogical tool to develop children’s social, play, and physical skills. They referred to recess time as “play lessons”. One teacher described the school’s approach to play: “Everything we do throughout the day, we’re prompting the kids, and our playground time is an actual programmed lesson, play lessons, where we’re teaching social foundation skills” (Teacher A2).

On the playground, children had an adult play partner (generally a teacher or therapist). Play partners guided children to meet specific, measurable goals related to play (e.g., throwing a ball or sharing toys with a peer). Staff at School A demonstrated a belief that children with ASD could and should be taught to play “properly”. The occupational therapist gave an example of play lessons:


*If I knew that Jack has a turn-taking outcome, I’d be like, “oh, Jack, let’s go on the trampoline, oh, it’s blah-blah-blah’s turn, okay, that means we have to wait”. And you’d try and facilitate those interactions that are working towards their goals, instead of just letting them work it out for themselves. (OT A1)*


The need for play lessons, according to the staff of School A, stemmed from disability: without specific teaching, children with ASD would not play “appropriately” (or, at least, they would not play as the staff expected children to play). One teacher explained, “I guess kids with autism, most of our kids, if we left them to their own devices, they’d sort of be quite happy in a corner on their own” (Teacher A3). Thus, staff felt that adult intervention was imperative during recess.

#### 3.1.3. Keeping Control over Risk and Challenge

The staff at School A recognized the need for children to experience risks and challenges, but many believed that these opportunities should be designed and monitored by adults. One teacher explained how she created situations for children to learn to ask for help: “With [learning to ask for] help, we’ve left them a bit longer…So, we’ve had a focus on waiting, teaching them what to do and then sabotaging situations so that they ask for help” (Teacher A1). Another teacher selected opportunities to step back, but only when she knew students would be successful:


*Although it might not be risk as such, you know, I step back a lot and let the kids try and problem solve, especially things I know they can figure out, well, they can eventually figure out, that’s not going to get them too upset. (Teacher A3)*


Organic risk-taking opportunities, such as climbing or solving conflicts, were tightly controlled and only allowed for certain students. Staff used their knowledge about individual children to decide whether they would permit risk-taking. A teacher explained why she allows some children to climb on play structures, but prevents others:


*Kids who climb all the time and I see from assessing them that they’ve got really great gross motor skills, I’d be more likely to step back and let them climb to the top of the tree, than I would a kid who trips over all the time and spends most of their time on the floor or falling over or has less skills. (Teacher A1)*


Staff reported two common reasons for stifling risky play opportunities. First, accountability to parents: many staff suggested that they might be more inclined to permit risks if they did not fear reporting injuries to parents. One teacher reported, “It’s probably the thing I fear most, having to ring a parent that a student has been hurt, especially when it could have been avoided if someone had just stepped in” (Teacher A3). Staff also attributed their low tolerance for risky play to behaviors associated with children’s autism—particularly “meltdowns” (Teacher A3) or tantrums resulting from an injury or challenge. The occupational therapist explained his approach to risk, “So we definitely think, because of our kids having some pretty intense behaviors, and they come on really fast a lot of the time, we tend to be really like risk managing, which is a terrible way to put it”.

#### 3.1.4. Theoretical Proposition: What Happened at School A?

Figure 3 is a visual representation of our hypothesized relationship between the cultures at School A, the assumptions underlying the intervention, and the outcomes of the intervention. Notably, this figure (and the narrative that follows) should not be interpreted as causative or definitive—myriad other factors likely influenced the intervention at School A. However, our theoretical proposition demonstrates a potential relationship among these factors, supported by our qualitative findings.

The cultures around play and disability at School A diverged significantly from the assumptions of the intervention. While we expected that adults who worked with children with disabilities would do so from a strengths-based perspective, the staff at School A foregrounded children’s disabilities. Moreover, we expected that staff would be motivated to allow children to engage in play for its own sake and to play with their peers; instead, we found that the staff viewed play as an adult-led, goal-oriented activity.

We propose that the mismatch between the cultures around disability and play/recess at School A and the assumptions underlying the intervention led to hesitance among staff members to step back on the playground. The pedagogical approach to play left staff unmotivated to attempt the intervention, favoring their existing “play lessons” approach. In an attempt to remain adherent to the intervention though, the staff largely avoided the loose materials:


*We always try and engage with the kids in the playground, and we didn’t stop trying to engage with the kids in the playground in the other areas, it was just in that area that was like the dead spot where you don’t engage with the kids. (Teacher A1)*


As a result, one teacher suggested the children did not enthusiastically approach the materials:


*I think the big thing that came up for me was in the playground we’ve been told and taught to play with the kids and interact with the kids, but we were told not to encourage them to play with the playground stuff. But somehow within that message, it kind of became if they’re playing with the playground stuff, walk away. So, I didn’t always do that, like I was sometimes playing with them when they started playing with it. But I think because they are so used to taking their lead by us, it ended up… didn’t get used that much. (Teacher A4)*


Additionally, the disability-first culture may have contributed to staff hesitance. The staff members overwhelmingly suggested that the children had too many limitations to benefit from this type of intervention. The occupational therapist, for example, foregrounded the children’s difficulties to explain the lack of engagement with the intervention materials:


*I just think, like, because these kids find it hard to play, a lot of them, I think perhaps if it had been presented a bit better, like if some materials had been put into the sandpit, if some had been put into the ball pit, if they’d sort of been spread out a bit. And maybe if the staff had been allowed to model, like, different things you could with it as well, I wonder if it would have got them playing a bit more. (OT A1)*


The children followed the staff’s lead and avoided the materials and little meaningful change occurred as a result of the intervention. Without seeing early positive results, the staff settled into their typical ways of engaging with children on the playground. Our team observed little change in the staff culture around risk, and children demonstrated few changes on the outcome measures.

Unexpectedly, however, seeing the poor results may have spurred cultural change at School A. During our interviews, which were conducted after the staff received preliminary results of the intervention, staff members reflected on the need to change their approach to risk after seeing the results.


*And so I really don’t think, upon reflection, we do give them enough of a chance to initiate the play because we’re always trying to get so much out of them, which is probably why they didn’t get as involved as what we would have thought they would. (Teacher A2)*


The occupational therapist and teachers suggested ways they might approach play differently after the intervention. One teacher proposed bringing her students to a nearby mainstream school playground to engage with other children instead of playing with adults; another suggested taking days off from play lessons and allowing the children to play independently. Unfortunately, because the study period concluded, we do not know if the staff members remained faithful to these plans.

### 3.2. School B

Next, we present the results of the analyses related to School B.

#### 3.2.1. High Expectations for Children with Disabilities

In contrast to School A, staff at School B emphasized that, for the most part, students with disabilities were not very different from other students. Mainstream teachers noted that there were students with disabilities in every classroom, not just the support unit. As a result, they felt a sense of responsibility to all students with disabilities: “We’ve got disabilities in every single classroom. So, if we’re not accepting of the disabilities within those units, then you know, that’s, I would just find that very um, hypocritical” (MS Teacher B2). Another teacher considered whether she treated students with disabilities differently on the playground:


*I don’t think I think of them any differently. They play the same games that other kids play, and they do the same thing that other kids, so not really. No. I let them do their own thing and if they look like they’re going to, you know, get injured or hurt someone, of course I’ll step in, but I don’t think I... No, I let them do their thing. Why not? We’re all entitled to it. (MS Teacher B4)*


Staff who worked in the special education classrooms (i.e., “support unit” staff), on the other hand, acknowledged differences between their students and mainstream students. However, throughout the interviews, their statements reflected high expectations for their students: “We’re always working towards—with high expectations or, ‘Of course you can do that; of course, you can do that because other kids can do that.’” (SU Teacher B1). One teacher felt that students with disabilities often just needed more opportunities to practice play and social skills: “Our kids need more and more chance to do it [practice play skills] and then they do learn”. Throughout the interviews, support unit staff acknowledged children’s limitations, but emphasized their abilities.

#### 3.2.2. Inconsistent Beliefs about Recess and Play

Staff at School B lacked a coherent philosophy about play and recess. Staff who taught mainstream students generally expressed that recess provided students with a break from the demands of the classroom. One teacher described recess as a “brain break” (MS Teacher B3). She emphasized children’s need for time away from the demands of the classroom. Another teacher agreed that recess was a break; however, when probed, she described organic learning opportunities on the playground:


*They, they’re learning how to, um, create friendships and learning how to maintain friendships. They’re learning how to play safely, um, abiding by their own rules of the games that they’re playing. Um, they’re learning to be you know, good sportsmen. (MS Teacher B2)*


Some staff, particularly those in the support unit, felt that the playground was primarily a place for children to learn. The SLSO emphasized her role in teaching children the physical skills she felt were necessary for peer play:


*One of my things, you know, I will get out there sometimes and play captain ball or tunnel ball or some of, you know, the catching games with my class and if other kids want to join in, fine you know, so be it, but my point as well is, our kids who haven’t got those skills, need to get those skills before they go and put themselves in a situation, where they ask other kids, “Can I play?” (SLSO B1)*


Across our interviews, opinions about the value of play and recess varied widely. This finding aligns with our observations of teachers’ practices on the playground. Some teachers primarily wandered around the playground, intervening only when they felt that student safety was at risk. Other teachers became involved in play, encouraging and engaging with the children.

While the perspectives surrounding play differed across staff members, both administrative leaders we interviewed felt that playtime was critical for children of all abilities. The deputy principal expressed the value of play. “I thoroughly believe that play is, basically for every aspect, is an important [part of] development, and it’s an important way that children cope with the world around them. And how they grow confident in the world around them”.

#### 3.2.3. Preventing Risk before It Happens

Staff at School B emphasized the need to prevent risky situations before they became too risky. Prior to the intervention, children on the playground at School B had few opportunities for physical risk-taking. The wide, grassy field did not offer natural occasions for risk-taking. The main exposure to risk in play was managing interpersonal conflicts. Staff emphasized their role in preventing conflict before it escalated. One teacher described her role: “Wander around, make sure, you know, no one’s hurting anyone. Make sure if there’s any issues that you note it down... I guess just standard rules” (MS Teacher B1). Another described growing conflicts as “hot spots”:


*We’re just ensuring that, um, there’s no hot spots happening and, um, the teachers that are, that are on duty are, you know, walking where they need to walk and stopping those little hot spots from you know, becoming enraged fires. (MS Teacher B2)*


Rather than providing children with the opportunity to resolve conflicts, staff often intervened to prevent them.

Like School A, School B staff emphasized their accountability to parents in keeping their children safe. A mainstream teacher explained, “It’s more safety. Make sure you’re doing things safely, so you don’t go home injured. We don’t want to have to call your parents and say, ‘They’ve done something silly on the playground’” (MS Teacher B3). A support unit teacher described herself as “hyper-vigilant” to avoid risks that she would have to report to parents (SU Teacher B2).

Leadership stressed risk-aversion as a significant problem that impacted staff and children. The deputy principal described the staff as “behind” in allowing children to manage risks: “Our risk taking is… there’s quite a literal response to risk management here in that, um, the only way to, um, help somebody change is to, sort of, lock them down or limit them” (Deputy Principal B1). The principal emphasized the impact of risk-aversion on the children:


*Traditionally the students have been quite compliant. Um, they’ve done what they’ve been told to do, kind of, but not so good at thinking for themselves and working things out. They’re very sort of rule-based and so they haven’t really worked things out for themselves. (Principal B1)*


Both administrative leaders we interviewed stressed the importance of the intervention to help the staff learn to accept more risk in play for children with and without disabilities.

#### 3.2.4. Theoretical Proposition: What Happened at School B?

Figure 4 is a visual representation of our proposed relationships among staff culture, assumptions of the intervention, and the intervention process/outcomes at School B. As was the case with School A, we intend this theoretical proposition to present a potential relationship, rather than a causative or definitive explanation for the study results.

The staff at School B shared our assumptions about disability—they had high expectations of their students. While staff did not explicitly share our assumptions about play, the culture around play and recess was marked by ambivalence and ambiguity. Most saw recess as a break and play as incidental, while a few felt that play was essential to children’s learning and wellbeing.

Compared to School A, at School B, we found relative congruence between the assumptions of the intervention and the culture. Perhaps, as a result of their high expectations for children with disabilities, staff were motivated to attempt the intervention. Despite some discomfort, staff members largely stepped back on the playground and allowed some risky play. A teacher recalled her reaction the first time she saw children wielding pool noodles as if they were weapons:


*I stepped away and I’m like, “You know what I’m gonna- I’m gonna wait until I actually- I genuinely think something’s there”. So, I’d wait over there and have my heart palpitating going, “Oh my gosh. Oh my gosh”. (laughs) But um, yeah, until like I actually took that step back, I’d be- like, you know, it took me a while to realize oh, that’s how they play. (MS Teacher B1)*


The children approached the materials enthusiastically. At School B, the ambivalent approach to play may have been buoyed by a relatively empty playground. Before the materials came onto the playground, the children had few opportunities to play aside from organized sports. The deputy principal recalled the children’s first time seeing the play materials:


*More than excitement, the absolute passion and glee that a large majority of the children attacked the, um, equipment, was phenomenal. They just saw it and couldn’t wait to get a piece of it. And, so, um, they, they were passionate. They were desperate. They were like, um, addicts I think, you know, just returning to their source of, um, uh, addiction, after a long break.*


In this case, it seems that the absence of a culture around play actually encouraged the children to approach the materials.

Because of the children’s enthusiastic uptake of the materials, the staff had ample opportunity to witness early positive impacts of the intervention. One teacher from the support unit observed the impacts of the intervention on her students with disabilities:


*So, what the playground project did for that, that- say that child with autism who, who plays by himself, is he had to play with others. And then those who were continually- they’re those who continually look for external sources of play, they then had to go up there and do imaginative stuff, because there aren’t- it was, it was junk, right?*


Having seen these positive results, many staff members continued to step back on the playground and encourage the students to play with the materials. The children continued to experience the benefits of the intervention and the staff’s culture around risk shifted to more risk acceptance.

An additional factor emerged throughout our interviews that likely impacted the intervention at School B—the engaged leadership team. We interviewed four members of the leadership team—the principal, the deputy principal, and two members of the teaching staff (one support unit teacher and one mainstream teacher) who held leadership roles. Each of these four staff members demonstrated enthusiastic support for the intervention and willingness to support the staff they supervised. The leadership team served as a liaison between the staff and the research team and held additional meetings with staff to address policy concerns related to the intervention. Their support was essential to the success of the intervention.

## 4. Discussion

In this qualitative multiple case study, we explored the impacts of widely held beliefs and practices on the implementation of a novel, school-based intervention. In the preceding section, we described the influences of staff cultures around play/recess, disability, and risk on the way that, and the extent to which, school staff engaged with the intervention. We achieved theoretical replication through the use of multiple cases [26]. Theoretical replication occurs when two or more cases “predict contrasting results but for anticipatable reasons” [26] (p. 54). In other words, the results of each case study may be different—but the lesson learned from each case is the same. In this study, the cross-case syntheses elucidated one fundamental lesson: existing staff cultures, and the extent to which these cultures aligned with the core assumptions of the intervention, influenced the implementation and subsequent outcomes of the intervention.

### 4.1. Differing Expectations: Influences of Culture around Disability on the Intervention

At School A and School B, the staff cultures surrounding disability were remarkably different. The staff at School A foregrounded disability when they spoke about their students. Their comments revealed a belief that their students were inherently limited. The staff at School B, on the other hand, acknowledged disability but did not place the same emphasis on it. Many adults, especially those who worked directly with students with disabilities, engendered a culture of high expectations. The school’s culture of high expectations may have contributed to the children’s ready engagement with the materials. When adults hold high expectations for children and adolescents with disabilities, they are more likely to succeed in a variety of domains including classroom behavior and obtaining post-secondary education [29,30,31]. The staff at School B, who believed students could benefit from the intervention, encouraged the children to use the materials and praised them for doing so. The staff at School A, expecting that children would not be able to engage in more sophisticated play with the materials, may not have encouraged them to do so.

While our interviews revealed much about the cultures around disability at each school, the reasons for the differences in staff culture remain somewhat elusive. One possible explanation for the differences lies in the nature of the two schools: a private specialized school (School A) and a mainstream public school (School B). Staff at School A had a much stronger focus on autism, while staff at School B taught children with multiple disabilities. It is possible that the staff at School A had more specific knowledge about the children’s disabilities; therefore, they were more inclined to attribute children’s characteristics and preferences to their diagnoses. At School B, on the other hand, staff were exposed to a greater variety of students with disabilities. Furthermore, although the special education classrooms had dedicated teachers, these teachers were often exposed to mainstream students. Their beliefs about what children can and cannot do, then, were shaped by children of all abilities.

### 4.2. What’s the Point of Play? Influences of Culture around Play/Recess on the Intervention

Prevailing beliefs and practices surrounding recess and play impacted the intervention at each school differently. The staff at School A held onto their pedagogical approach to play, maintaining a high level of engagement with children on all parts of the playground except where intervention materials were located. At School B, on the other hand, there was no prevailing culture surrounding play or recess. The intervention provided a guidepost for promoting child-led play, and the staff members generally accepted the intervention.

Sutton-Smith [23] described seven rhetorics (i.e., dominant narratives) that adults use to conceptualize play. Staff at School A subscribed primarily to the rhetoric of play as progress, which foregrounds the developmental benefits of play. Peterson and colleagues [24] reviewed the rhetorics of play implicit in kindergarten curricula. Not surprisingly, and like our findings in School A, play as progress dominated. Peterson et al. cautioned against instrumentalizing “purposeful” play, warning that this narrative could lead to the demise of free play, which seems to characterize what we saw in School A.

### 4.3. Disability and Play: Intertwined Influences on the Intervention

While the cultures around play and disability each separately influenced the intervention, their combined impact may have been even greater. At School A, staff held a prevailing sentiment that because of disability, children needed adult support to play. There is little doubt that children with ASD—and some children with ID—play differently than other children. Children with ASD often demonstrate less frequent and less sophisticated social and imaginary play than their peers [32,33]. Correspondingly, a great deal of the literature focuses on developing ways for adults to teach children how to play [34,35]. Much of this literature is predicated on the unsubstantiated assumption that children with ASD will be better or happier if their play more closely resembles that of typically developing children [36], a belief that School A staff also seemed to hold. Unfortunately, such approaches may rob play of the very components that make it play: intrinsic motivation and internal control. School B staff also may have held that belief but their approach to achieving it was very different from that of School A staff. In integrating the children from the support units with the mainstream children, they may have hoped that the children in the support unit would emulate the mainstream children in their play.

Our intervention took a novel approach to promoting play for children with ASD and other disabilities. Instead of an adult-directed approach, we offered children loose materials with no obvious play value. Because many of these materials were large and/or heavy, the children had ample motivation to engage in cooperative or social play. Furthermore, because the materials had no prescriptive play purposes, children were free to use their imagination. Unfortunately, this approach fundamentally did not align with the widely held beliefs about play and disability at School A. Interventions that are misaligned with existing values, priorities, and school policies are less likely to be implemented successfully and sustainably [18,37]; this was very likely the case at School A. In contrast, these materials may have assisted children in School B to play more like their typically developing peers.

### 4.4. What about Risk?

The influence of cultures around risk on the intervention is decidedly less clear. At the start of the intervention, staff at both schools demonstrated aversion to risk. Of course, this is what we anticipated—and indeed, what we attempted to address through the RRWs. However, their definitions of risk differed: Staff at School A defined activities as risky if they might lead to injury or “meltdown”; Staff at School B focused on interpersonal risks, such as arguments or physical fights among children. Little research has examined adults’ definitions of risk or risky play. Little, Sandseter, and Wyver [38] interviewed early childhood teachers in Australia and Norway and found that definitions of risky play differed between the two countries. However, we identified no studies examining risk definitions within or between schools in the same country.

Another element of the culture around risk that separated the two schools was the influence of leadership. At School B, members of the leadership team were particularly supportive of risky play, while at School A, administrative leaders had little direct involvement in the intervention. Support from administrative leadership is a critical piece of implementing and sustaining a novel intervention [18,39,40]. School principals and other leaders play a pivotal role in establishing, and, if necessary, shifting staff culture [39]. Leadership may have been particularly important in overcoming a culture of risk-aversion. Staff from both schools feared retribution from parents if a child were injured on the playground. Complaints about teachers’ duty of care are typically channeled through administrative leaders. Staff at School B felt that leadership would support them if a child got hurt—this did not seem to be the case at School A. Possibly, the direct support from leadership at School B, together with the intervention, overrode the risk-averse culture.

### 4.5. Limitations

Of course, we cannot make definitive claims about the sources of differences in either implementation or quantitative outcomes. Myriad factors may have impacted the study. School B housed mainstream students, while School A did not. Conceivably, the mainstream children demonstrated and encouraged play for the support unit students. Typically developing peers have often been used as models in interventions targeting play for children with disabilities, particularly ASD [40]. Additionally, we have no diagnostic data about the children who participated in the study—it is possible that the children at School A had more severe disabilities than those at School B. Staff at School A believed this was the case. Previous studies suggest that disability severity impacts play and playfulness [41,42].

Additionally, our findings are limited by the absence of implementation-level outcomes, such as the feasibility, acceptability, or sustainability of the intervention [43]. Because this was a retrospective study, we examined implementation through the data available to us: evaluative interviews with staff involved in the intervention. While this allowed us to develop a general narrative of implementation at each school, our understanding of each schools’ implementation processes may be limited. On the other hand, several authors of this study were deeply embedded in the schools during this study; their perspectives served to strengthen the trustworthiness of our findings.

Notably, we had more interview participants from School B compared to School A. At the time of data collection, we did not intend to compare schools—therefore, we did not attempt to match these two groups of participants. Furthermore, we had a wider range of participant roles at School B (likely reflecting the greater involvement of leadership at this school). We acknowledge this as a potential weakness in the study design.

Finally, case study methodology is inherently limited in its generalizability. We cannot say with certainty that the results of this study will be indicative of future schools. However, the cross-case synthesis approach supports our findings [26]. Yin compared cross-case syntheses to replication studies. Although the outcomes were different at the two schools, the conclusion of each case study is the same: staff culture around play/recess, disability, and risk influenced the implementation and results of our intervention.

## 5. Conclusions

### What Could We Have Done Differently?

When we initiated the playground intervention, we aimed to shift staff cultures around risky play. However, the adult-directed program we developed mainly addressed culture around risk—we did not address culture around play or disability. We assumed—incorrectly—that school staff shared the values of the intervention team. The results of this study highlight the importance of researchers examining their own fundamental assumptions as well as coming to a deep understanding of existing staff cultures before initiating a novel intervention and tailoring the intervention to address the multidimensional, complex cultures that exist in schools. Had we done so, we may have made several adjustments for School A. We may have challenged staff to examine their beliefs about play and disability during the RRWs. We may have recruited administrative leadership to champion the intervention. Because the playground intervention was part of a cluster trial, we aimed to keep the intervention as similar as possible across schools; therefore, we did not take these steps. However, the results of this study illuminate the need for school-based researchers and practitioners interested in promoting risky play to take careful stock of each unique school environment.

## Figures and Tables

**Figure 1 ijerph-18-01008-f001:**
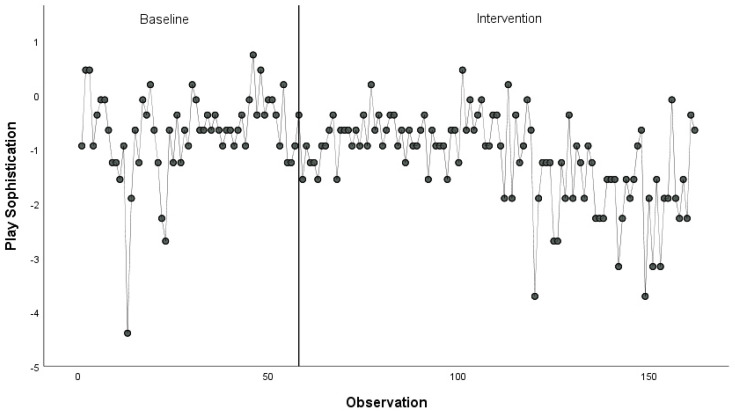
Playground observation scores during baseline and intervention at School A.

**Figure 2 ijerph-18-01008-f002:**
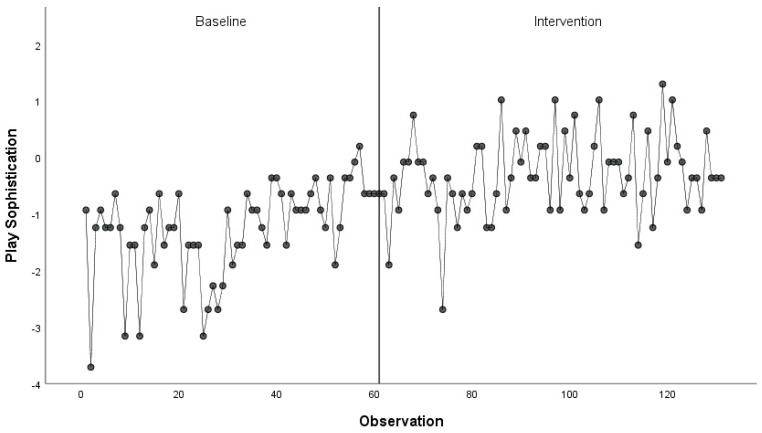
Playground observation scores during baseline and intervention at School B.

**Figure 3 ijerph-18-01008-f003:**
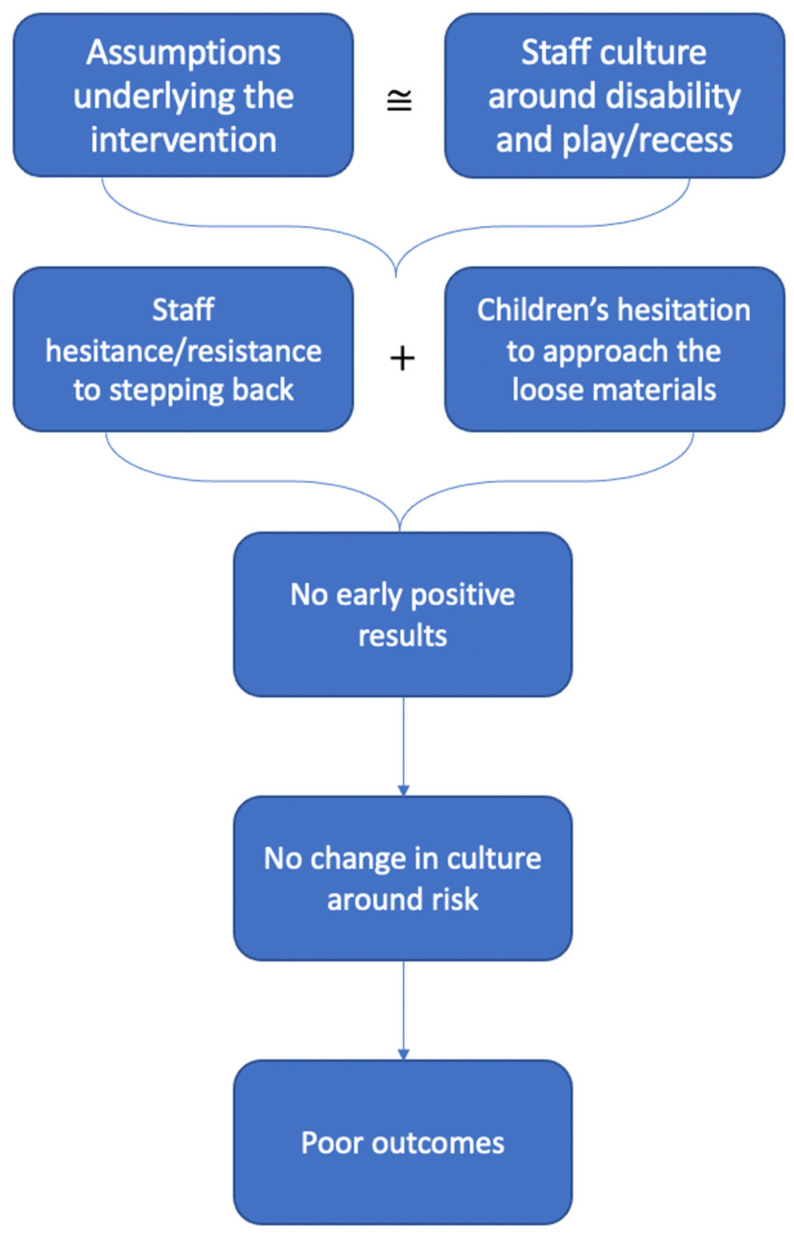
Proposed relationship among aspects of staff culture, assumptions underlying the intervention, and the intervention at School A. ≅ refers to congruence.

**Figure 4 ijerph-18-01008-f004:**
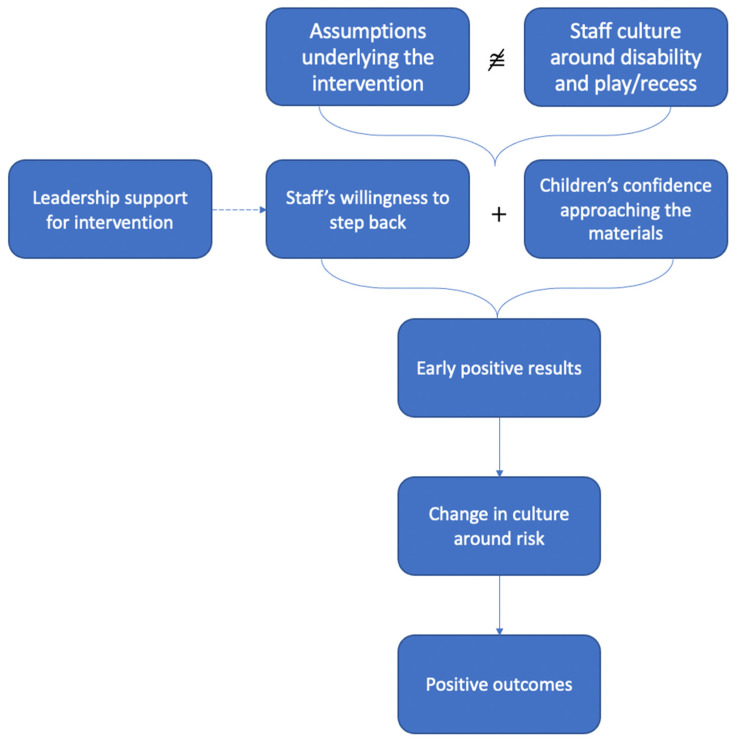
Proposed relationship among aspects of staff culture, assumptions underlying the intervention, and the intervention at School B. ≅ refers to incongruence.

**Table 1 ijerph-18-01008-t001:** Quantitative outcome measures and analyses.

Outcome	Unit of Analysis	Data Collection	Analysis
Sophistication of Play Activities	Playground session	Systematic observation using study-designed, iPad-based application [14]. Trained observers collected data during outdoor recess 3–5 days/week during control and intervention phases. Observers coded play categories (e.g., rough and tumble, sensory play).	Rasch analysis to construct interval-level sophistication scores for each play session based on play categories. Visual inspection of scores over time. Descriptive comparison of mean scores for control and intervention phases.
Play Engagement	Playground session	Systematic observation using iPad-based application described previously. Observers coded each observation of play as (4) very engaged, (3) more engaged than not, (2) somewhat engaged, or (1) minimally engaged.	Dichotomized scores into more engaged (3 or 4) and less engaged (1 or 2). 2 × 2 χ^2^ to compare control and intervention phase counts. Φ to evaluate effect size [15]: ○ 0.1 = small effect; ○ 0.3 = medium effect; ○ 0.5 = large effect.
Coping Ability	Individual child	Staff member familiar with target children completed Coping Inventory [16] during control and post-intervention.	Paired *t*-tests to compare control and post-intervention scores. Cohen’s *d* to evaluate effect size [15]: ○ 0.3 = small effect; ○ 0.5 = medium effect; ○ 0.8 = large effect.

**Table 2 ijerph-18-01008-t002:** Interview participants, pseudonyms, and roles.

	Pseudonym	Role
School A	OT A1	Occupational therapist
	Teacher A1	Classroom teacher
	Teacher A2	Classroom teacher
	Teacher A3	Classroom teacher
	Teacher A4	Classroom teacher
School B	Principal B1	Principal
	Deputy Principal B1	Deputy Principal
	SU Teacher B1	Support Unit Teacher
	SU Teacher B2	Support Unit Teacher
	SLSO B1	Student Learning Support Officer (working directly with students from the support unit)
	MS Teacher B1	English as an Additional Language Teacher
	MS Teacher B2	Mainstream Teacher
	MS Teacher B3	Mainstream Teacher
	MS Teacher B4	Mainstream Teacher

## Data Availability

The data presented in this study are available on request from the corresponding author. The data are not publicly available to protect confidentiality of the participating schools and staff members.

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
