# Peer review of "Expectations and Assumptions: Examining the Influence of Staff Culture on a Novel School-Based Intervention to Enable Risky Play for Children with Disabilities"

_ijerph, 2021, doi:10.3390/ijerph18031008_

Round 1

Reviewer 1 Report

Thank you for allowing me to review this interesting study. Overall, the study has raised a very interesting discussion point about the cultural beliefs about disability and play in children with special needs. I believe this study has provided novel findings in this area, allowing readers to think deeper and more thoroughly when they promote play in children with special needs to their parents at home and teachers at schools.

There are a couple of minor points if the authors may amend or clarify:

  1. page 8-  the heading of 3.1 should be School B, not School A.
  2. page 15, lines 587-589- Italic words are used for the direct quotes from the teachers. Here, I am not sure if this is a quote from the study teacher or an argument from the authors. Please clarify.

Overall, it is a very interesting manuscript. However, as this is a qualitative study, I believe the authors may want to tune down the study findings in general. As commended under 5.5 Limitations by the authors, there are numerous factors may affect the findings and with the limitation of the present study design; hence, no strong definitive causative relationship should be drawn between the teacher culture in disability to enable (risky) child-directed play in children with special needs.

Reviewer 2 Report

The paper presents the results of a retrospective study in which the implementation process of an educational intervention aimed at enabling risky play for children with disabilities is compared in two different schools.
The theme of the paper is highly innovative and may be of interest to potential readers of the article.
The first part of the paper summarizes the main results of the intervention carried out. And the second part analyzes, from a qualitative point of view, some of the reasons that may explain the differences between two schools in the implementation of the intervention.
The qualitative methodology adopted presents some risks, derived from the involvement of some of the researchers with the schools and with the intervention itself. However, sufficient measures have been taken to ensure the trustworthiness of the investigation.

There are some issues of the paper that the authors should reconsider or at least take into account:

1. Lines 123-124.
"However, in 2018, 31.2% students with disabilities attended either specialized schools or segregated classrooms"
It must be specified which country this data is from. Furthermore, given that the potential readers of the paper are international, more data should be provided from other educational systems on the enrollment rate in ordinary schools for students with disabilities.

2. Research question 1.
"What was the staff culture (i.e., shared beliefs, norms, values, and practices) surrounding disability, play/recess, and risk-taking that characterized each of the two schools?"
Before this research question it must be specified that two schools are studied. This is stated on the abstract, but not on the text.

3. Text structure.
3.1. Table 1.
I'm not sure if table 1 should be located in the introduction.
This table provides information necessary to understand the results of the interventions. Authors are encouraged to consider relocating this table to the Methods section. Although this information does not belong properly to the method of the present study (which is based on the analysis of interviews), it is necessary to understand the results of the intervention.
3.2. Description of schools.
I think a more logical structure would be to first introduce the schools and then describe the professionals who were interviewed. In other words, the "school A" and "school B" sections must be placed before the "Interview Participants" section.

4. Main conclusion of the paper.
The authors attribute to the school culture the differences between the professionals of the two schools in the approach to play in recess time.
I agree with the authors that this variable is highly relevant.
However, have the authors analyzed if the degree of severity of disability in both schools is the same?
That is, as authors mention on lines 701-703, it is possible that school A (a specific special education school, with a lower ratio of students per classroom), only admits students with more severe disabilities than school B.
I mean, it is possible that the precautions of the professionals of school A (and the school culture itself) are due to the fact that the degree of disability of their students is significantly greater than that of the students of school B.
From my point of view, the major limitation of the study is that the characteristics of the children in both schools are not fully controlled. Therefore, this issue should be explained by the authors in greater depth, if possible, stating the criteria that are followed in the educational system of which the schools are part to enroll a student in a school type "school A" or in a "school B" type school.
Otherwise, the readers of the paper could (with good reason) assume that the differences are not found in the school culture, but in the children who are the final recipients of the intervention.

5. Reasons for the differences in the school culture of the schools.
Once the existence of differences in the school cultures of the schools has been verified, it is necessary to seek explanations for these differences.
One possibility is that these differences derive from the initial training or the selection process of the professionals from both schools.
Regarding initial training, is it possible that the professionals from school A have a more clinical training, more focused on the analysis of disabilities, and that the professionals from school B have a more pedagogical training?
Is it possible that there are differences in the selection of teachers from both schools that include biases in school A that could favor the results obtained in this study?
Authors are encouraged to try to explain in greater depth what the causes of differences in school cultures are, rather than just assuming that these differences exist.

6. Formal issues.
6.1. Keywords must be in alphabetic order.
6.2. When "p", "d" and "phi" values are provided, the "0." part of the number should not be indicated. That is, p = 0.001 should not be written; p = .001 should be written.
6.3. Line 275. Erratum. Is it "3.2. School B"?
6.4. All section and subsection numbering should be reviewed. For example, currently the method section has the number 2; and its subsection "case selection" is numbered "3.1".

Reviewer 3 Report

First of all I would like to share the need to carry out work like the one you present. They are necessary for the advancement of science in the field of disability and gambling. Play is the basis in the early stages of life and in the field of disability it takes on an important pedagogical value.

This study aims to understand and modify the culture of the staff (that is, shared beliefs, values ​​and practices around the risk game in order to carry out an objective intervention. The study has been an interesting read, however I have some questions and suggestions for authors.

It is necessary to know the existing relationships between competitive and risk gambling, since it is basic and fundamental to address capacities and limitations.

I find the intervention carried out both to parents or legal guardians, and to children, really interesting. These are aspects that must go hand in hand, especially when the area of ​​disability is the one in the middle.
I would like to show that the instrument used in the intervention is strongly supported by the scientific community and provides objective, contrasted data to be able to be replicated.
How was this awareness of parents or legal guardians carried out? Because I understand that they have had a conversation with those in charge of the centers to choose them, through a semi-structured interview. Based on the results, orient the talks in the schools? Was that talk the same in all the schools? Was it done by the same researcher? or it was several members who carried out this work.
Don't you think that participation in a school of 9 and in another of 5 people may have influenced the results? I would like this appreciation to consist of work limitations. I would like to know if the ideology of the centers was religious or not. And if so, if the author considers that this could be a limiting factor or, on the contrary, it helps to promote this type of practice. I would like it to be included in the manuscript if that is the case. I also invite you to continue along this interesting line.
